# How Does Targeted Poverty Alleviation Policy Influence Residents' Perceptions of Rural Living Conditions? A Study of 16 Villages in Gansu Province, Northwest China

**Meimei Wang [1], Yongchun Yang [1,*], Bo Zhang [2,3], Mengqin Liu [4] and Qing Liu [1]**

[1] College of Earth and Environmental Sciences, Lanzhou University, Lanzhou 730000, China; wangmm@lzu.edu.cn (M.W.); liuqinglwz@163.com (Q.L.)

[2] School of Tourism Management, South China Normal University, Guangzhou 510631, China; bo.zhang@rug.nl

[3] Faculty of Spatial Sciences, University of Groningen, 9700 Groningen, The Netherlands

[4] College of Environment and Resource, Southwest University of Science and Technology, Sichuan 621010, China; lmq4336@163.com

[*] Correspondence: yangych@lzu.edu.cn

**Abstract:** Rural living conditions (RLCs) in China are influential on the overall development and stability of regions, particularly for populations in distant poverty-stricken villages. This paper takes 16 villages of Chedao town in Gansu province, Northwest China (NWC) as our case study. Using data from the Poverty Alleviation and Assistance (PAA) project launched by Lanzhou University in June 2017, and the perceptions of residents of Chedao, we pinpoint RLC changes in the targeted poverty alleviation (TPA) process. The three main results show that: (1) From the residents' perceptions, the impact of alleviation measures on RLC is mainly reflected in improved housing conditions, infrastructure, and public services. We find no significant effect on cultural conditions. However, eco-environmental conditions have obviously weakened. (2) Housing size, accessibility, distance to shops, and safe drinking water are the most significant factors in housing conditions, infrastructure, public services, and eco-environmental conditions, respectively. (3) Out of the different levels of rural poverty households (RPHs), severe rurality villages are more strongly aware of the positive changes in RLC than residents of mild rurality villages. Moreover, in residents' view, housing conditions are most improved in *severe* rurality villages, infrastructure is most improved in *moderate* rurality villages, and public services are most improved in *mild* rurality villages. Eco-environmental conditions worsen across all levels. Our findings shed light on the perceptions of residents on changes occurring in rural living conditions, and provide a basis for subsequent studies of RLC in Northwest China.

**Keywords:** absolute poverty; rural poverty household (RPH); targeted poverty alleviation (TPA); resident perceptions; rural living condition (RLC); Northwest China (NWC)

## 1. Introduction

On 26 August 2016, a poverty-stricken mother named Yang Gailan, of Gansu province, committed suicide by ingesting pesticide after killing her four under-aged children with an axe. Eight days later, on 4 September, the body of Yang's husband was found in a wooded area of the village; he was a victim of self-poisoning. The tragedy of this household sparked wide discussion on poverty issues in rural China. Consistent with the stance of Chinese authorities, the state media propaganda declaimed the nature of the incident as 'individual and one-off,' and admonished that Yang should have taken

responsibility for her own actions after several local officials had imposed disciplinary sanctions upon her. Overseas media thought otherwise, however, and the main criticism focused on poverty alleviation policy. The Yang household was widely known as one of the poorest in the village and had been receiving a targeted poverty alleviation allowance. However, this allowance was cancelled in 2014 by local authorities because the annual income of the Yang household was ¥4717.7/per person, which was above the standard for receiving the allowance (¥2300/year/per person). We can safely say that suffering from poverty must be at least one of the main reasons why the young mum took such desperate action.

Given its rapid economic growth and strong government intervention, China has made great strides towards reducing poverty since 1990. Estimates show that over 500 million Chinese citizens have been lifted out of extreme poverty (UNDP 2017). In the report of the 18th National Congress of the Communist Party of China (CPC) [1], China vowed to save all of its residents from poverty and provide them with a well-off society by 2020. When we compare poverty as equivalent to the dollar-based income standard of $405/year [2], we see that nearly 56 million rural citizens are still living in 'extreme poverty.' In fact, the tragic circumstances of the Yang household are a miniature representation of the millions of poor people in rural China. Rural families still lack basic living necessities such as safe drinking water, basic health care, electronic devices, and even habitable shelters. Their children may not be receiving quality education [3]. The construction of rural infrastructure also largely lags behind urban areas. Many villages in Northwest China (NWC) still lack access to the outside world, and many roads are unpaved or are poorly paved. Therefore, the task of tackling poverty remains formidable.

The systematic Poverty Alleviation Project was first launched in 1986 by the State Council's Leading Group Office for Poverty Alleviation and Development (SCLGOP). During this time, a systematic approach was taken to assess the number of poverty-stricken areas based on an annual per capita rural income: of ¥150; this number rose to ¥200 in minority areas and ¥300 in 'revolutionary based' (an area with armed forces and political power established by the communist party of China in certain areas) counties. In total, 331 county-level administrations were designated as poor areas. By 1994, when the central government introduced the '8–7 plan', the number of poor county-level administrations had risen to 592, with an adjusted poverty level at ¥700 Yuan/capita rural income. The poor counties became the focus for this poverty alleviation plan. Instead of directly subsidizing the targeted areas, however, this project was designed to foster long-term and sustainable development to improve the incomes and jobs of rural residents and also attend to basic infrastructure: to build roads and provide safe drinking water facilities [4]. The third wave of top down poverty alleviation was initiated between 2001 and 2010 due to a dramatic increase in poverty. The poverty alleviation project identified 148,000 poverty-stricken villages and announced that social participation and comprehensive development would be the core strategies to alleviate poverty, rather than focusing only on improving incomes.

The government-led poverty alleviation project entered into yet another new era from 2011. At this time, however, it became much more difficult to alleviate poverty because of the rising standard of the poverty line (increased to ¥3335 per person per year in 2017) in addition to higher cost-of-living expenses. In fact, the poverty line itself was denounced for undercounting the truly needy poor population because it only considered income [5]. Moreover, Chinese authorities also identified shortcomings of the previous project, such as some impoverished families living in non-designated poverty counties [5].

Poverty accelerated massive rural-to-urban migration, rural residential land has been increasingly abandoned and many houses were left in disrepair [6]. The understanding of poverty has gradually evolved from a pure monetary perspective to a more multidimensional view [7]. Chinese authorities therefore proposed the ambitious idea of targeted poverty alleviation (TPA) in 2013 (*Jin Zhun Fu Pin*), which highlights an accurate poverty identification, appropriate prioritization of projects, and efficient implementation to ensure that assistance reaches all of the poverty-stricken villages and households [7]. In TPA policy, rural dilapidated house transformation (*Wei Fang Gai Zao*) pays close attention to RLC and is indeed aimed at the housing problem.

In the present work, we examine the effectiveness of the TPA project launched in 16 villages in Chedao town, Gansu province. We have collected data from this area because the authors have conducted much of their fieldwork in Gansu, one of the most severely poverty-stricken provinces in China. It seems that the infrastructure was improved during our several visits to Chedao town, however it is still unknown how exactly the residents feel about these improvements. The perceptions of the residents can be important indicators to assess the present public policy against rural poverty. Thus, we set out three research questions as follows: What is the status quo of RLC of RPHs? How have RLCs changed during TPA? And finally, what are the differences in residents' perceptions with regard to changes in RLC among different groups? By answering these questions, we hope to clarify changes of residents' perceptions in RLC during TPA, evaluate the impact of TPA on RLC in different villages, and provide a scientific basis to adjust and improve China's TPA strategy.

The contributions of this article are threefold: 1. We focus on an area on which there is little previous research because of the low accessibility for scholars. 2. The existing literature mostly concentrates on how the improvement of infrastructure has influenced the livelihoods of the residents, while less research has paid attention to how the rural residents feel and what they are experience after changes in rural living conditions. Specifically, in this article, we stress on perceptions of the residents. 3. Although this study is conducted in a small scale, we find the degree of poverty still varies considerably in terms of the geographical distributions of these villages. The results indicate that the local government does need to make effort to provide targeted poverty alleviation for residents due to their geographical distributions.

## 2. Literature Review

In 1981 the World Bank defined poverty as the lack of opportunity for some groups who are without sufficient resources to purchase food, lack decent living conditions, and who have minimal participation in activities that are generally accepted by society. The definition of poverty was later simplified as the lack of the ability to achieve a minimum living standard [8]. Lipton (1984) divides poor people into 'ultra-poor and 'the poor' [9]. Chambers (1981) uses the terms of 'very poor' and 'poor' to describe poverty [10]. In this article we use the term 'poverty households' to avoid confusion with previous definitions; however, there are overlaps among them. The concept 'poverty households' was introduced in China's targeted poverty alleviation (TPA) process; it refers to poor households without special policy subsidies, and who have an annual per capita net income that is usually below the Chinese national poverty line (¥3335 in 2017) [11].

Poverty is not only reflected in low living standards, but also in poor living conditions. Rural living conditions (RLC) as more than a symbol of quality of life, as they can have various benefits for physical health and psychological welfare [12], and also contribute to subjective well-being and happiness [13]. According to Lee (2014), the settlement environment and rural environment affect life satisfaction; but at the same time, poor living conditions can threaten public welfare, and as a main cause of the depopulation of villages, can also reveal the types of solutions required to alleviate it [14]. According to scholars, environmental quality of life domains include satisfaction with housing, schools, health services, safety and security, roads, and access to transport [15–18]. In a rural study by Chow (2005) on assessing the degree of satisfaction degree with different aspects of life, respondents were most satisfied with living conditions and living arrangements [19]. In another study by Liu [20], the subjective well-being of rural residents was lower than for city residents because of inadequate social undertakings, poor water and electricity supplies, and insufficient wi-fi connectivity. In a study of health services in rural areas, Lin et al. (2011) cite dissatisfaction due to an insufficient number of clinics with poor medical services, shortage of medicines, and overcrowded facilities [21].

The RLC of RPHs has witnessed great changes in the TPA process, and research on the perceptions of poverty groups toward these changes has great practical significance. The existing works on residents' perceptions and RLCs focus on satisfaction and environmental quality. Further studies also indicate that people with better living conditions feel happier. The study by Chow confirms that those

who enjoy a high socio-economic status show high satisfaction with their living conditions [19]; Chow's finding is also consistent with recent research results [22–25]. Lee shows that, for elderly households, housing conditions are more important than eco-environmental conditions [26]. Despite mention of the studies above, research on RLC in China, especially in the impoverished villages of western China, has long been neglected compared to that of urban areas [27]. We stress that scant data and limited access to these northwestern villages has resulted in insufficient research of RLCs by TPA. Zhou et al. hold that the difficulties inherent in tackling poverty, fragile ecological environments, rapid population increases, aging, and rural decline pose major challenges to the development of a fully well-off Chinese society [28].

For the purposes of our study, we refer to the Multidimensional Poverty Index (MPI) for developing countries. Its micro datasets include education, health, standard of living, and other related dimensions [29–31]. Correspondingly, the Decade Poverty Reduction Programme (DPRP 2011) of China proposed improvements to residence, medical services, and education. In fact, these dimensions and aspects have been widely discussed in academia. Angulo et al. designed poverty reduction goals in Colombia by monitoring their public policies [32], while Rogan highlighted the missing gender dimension in poverty reduction policies [33]. The main findings of a study by Zhi et al. pointed to the need for improvements in education and nutrition, and the availability of cooking fuel and clean water as major factors related to multidimensional poverty [34]. Lastly, a case study of the Khanasser Valley in northwestern Syria by Rovere et al. found that agriculturists and laborers in rural areas with sufficient land indeed benefit from poverty reductions, but for the poorest households with little land, the works above make the evaluation index more diverse [35].

There are nevertheless some drawbacks to the aforementioned research. Three results in particular stand out: (1) the research results on the villages of extreme poverty in northwest China are sorely lacking; (2) it is not yet clear how living conditions have changed since the implementation of poverty alleviation measures in China; and (3) there is not yet a robust system framework on resident perceptions of a changed living environment.

## 3. Data and Research Setting

We conducted our research in 16 villages in Chedao town in NWC, situated in an area with an abundance of oil. The groundwater is non-potable due to its excessive salt and heavy metals; therefore, drinking water is mainly made available through and dependent on rainfall. However, rainfall in this arid zone is scarce. Transportation accessibility is also low due to the severe conditions of the roads and there is no railway system. The total number of households in Chedao town is 4818, of which 2203 (45.72%) are rural poverty households (RPHs) (NBSC 2017). The data in this article have been derived from the PAA (Poverty Alleviation and Assistance) project launched by Lanzhou University in Gansu province in June 2017, which aims to survey the problem of rural poverty in remote and backward villages in Gansu province. We have drawn from the two surveys on RLC in the PAA, which also define our categories *housing conditions*, *infrastructure*, *public services*, *eco-environmental conditions, and cultural conditions in perceptions of* rural poverty households (RPHs).

The samples in the PAA project meet the criteria of a rural poverty household (RPH): (1) The primary income provider of the household is under age 60; (2) Using the 2017 official poverty rate, per-capita income is below the government's official poverty line of ¥3335 ($530.12) per person per year.

During the TPA process, rural living conditions (RLC) have changed, but different perceptions exist because RPHs have different resources. It was therefore necessary to subdivide the study area based on rurality. We established the rurality index model based on the present studies [36] in conjunction with the rurality of 16 villages with 2203 RPHs. Table 1 sets out the index of rurality indicators, while the calculation of weights (Wi) is based on the variation coefficient method [37]. The extreme difference method is used in data standardization and the different directions of indexes are identified. According to index weight and standardized value, the rural indexes of each village are calculated. Villages with similar property values are set as a cluster by ArcGIS, the 16 villages in Chedao town are divided into

mild rurality villages, moderate rurality villages, and severe rurality villages (See Figure 1). Villages in Chedao town vary tremendously in terms of rurality. Villages with mild and moderate rurality are located in the northeast part of Chedao town where the agricultural productivity is relatively higher and small-scale cooperatives are relatively better established, therefore, fewer people move out for a living. In contrast, the villages with severe rurality are highly constrained by a shortage of water and low agricultural technology, in other words, the agricultural productivity in these villages is largely dependent on weather conditions. This means many people work outside the villages and the villages therefore face a labor shortage. The conditions in the villages with moderate rurality lie somewhere in between mild rurality villages and severe rurality villages.

**Table 1.** Rural evaluation indicators.

| Index of Rurality | Definition | $W_i$ |
|---|---|---|
| Change rate of cultivated land area | (Cultivated land area at end of year—that at start of year)/cultivated land area at start of year. | 0.130 |
| Change rate of rural population | (Rural population at end of year—rural population at start of year)/rural population at start of year). | 0.330 |
| Employment rate of primary industry | Labor of primary industry/total labor force. | 0.040 |
| Output rate of cultivated land area | Gross output value of agriculture/total cultivated area. | 0.120 |
| Per capita value of industrial output | Total industrial output value/total population. | 0.180 |
| Proportion of value of secondary and service industry | Value of secondary and service industry/GDP. | 0.190 |

We also designed a structural equation model (SEM) to test the causal path relationship between the latent variables. The relationships and predictions of variables were verified by the AMOS 19.0 and SPSS 20.0. Box plots were used to analyze differences in different groups.

Descriptive Statistics

Of the 2203 rural poverty households (RPHs), 1300 (59.01%) surveys were completed by men and 903 (40.99%) were completed by women. The age ranges 20–35, 36–45, and 46–60 years old accounted for 31%, 36%, and 33% of the respondents, respectively. The percentage of RPHs that live in mild rurality villages, moderate rurality villages and severe rurality villages is 31%, 31%, and 38%, respectively.

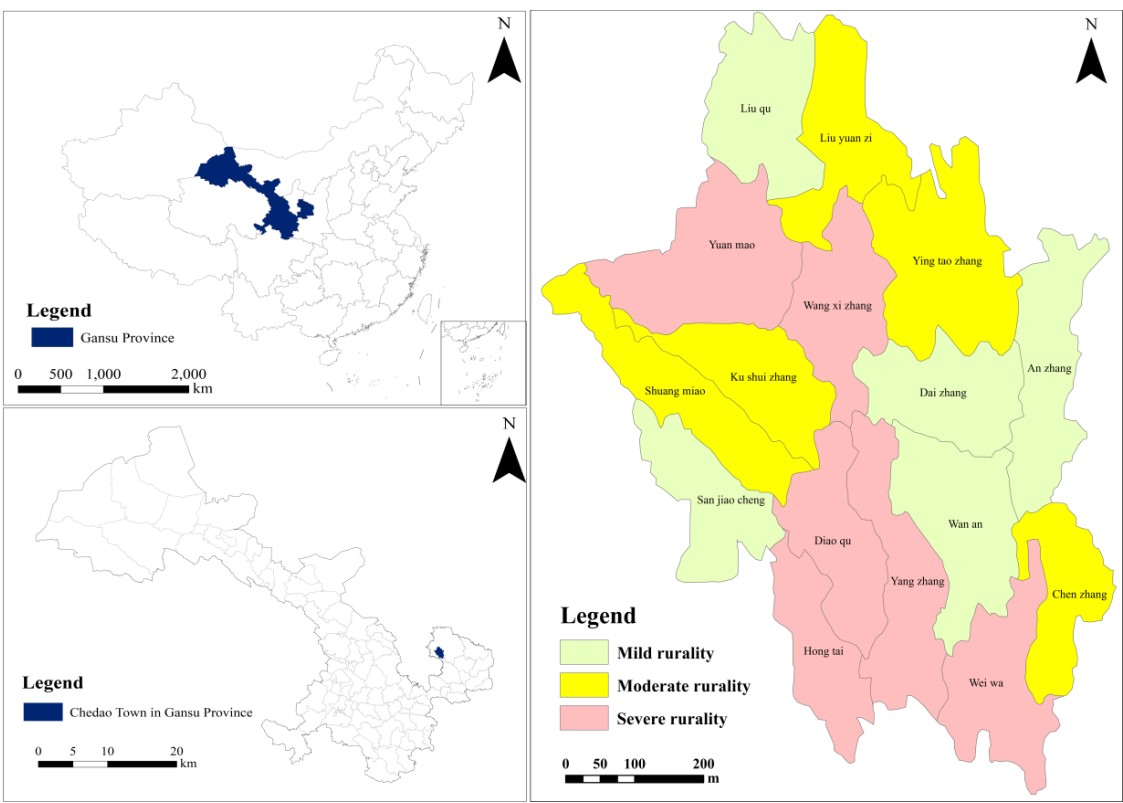

**Figure 1.** Rurality of villages in Chedao town.

In the PAA project, the RLC contains housing conditions, infrastructure, public services, eco-environmental conditions, and cultural conditions. These five aspects of the RLC are measured by the 26 questions shown in Table 2. We have used the 5-point Likert scale (from 1 to 5, with 5 being the highest value) in refinement of the analysis. Values of split-half reliability of housing conditions, infrastructure, public services, eco-environmental conditions, and cultural conditions are, respectively, higher than 0.5 and are thus significant. The value of Cronbach's Alpha in housing conditions, infrastructure, public services, eco-environmental conditions, and cultural conditions are between 0.35 to 0.7, indicating good quality of the observation data.

**Table 2.** The residents' perceptions index toward the changes in rural living conditions (RLCs).

| Latent Variables | | Measurable Variables | | |
|---|---|---|---|---|
| **Symbol Type** | | **Indexes** | | |
| **Exogenous Latent Variables** | | | **Mean** | **Stdev** |
| Housing conditions | $X_1$ | Housing size | 3.733 | 0.992 |
| | $X_2$ | Dilapidated walls | 2.100 | 0.876 |
| | $X_3$ | Housing type | 2.113 | 0.900 |
| | $X_4$ | Housing quality | 3.479 | 0.993 |
| | $X_5$ | Housing landscape | 2.154 | 0.945 |
| Infrastructure | $X_6$ | Water supply | 3.257 | 1.021 |
| | $X_7$ | Proximity to main road | 3.727 | 1.012 |
| | $X_8$ | Earthquake resistant capability | 2.071 | 0.847 |
| | $X_9$ | Fireproofing capability | 3.653 | 0.881 |
| | $X_{10}$ | Accessibility | 2.810 | 0.977 |
| | $X_{11}$ | Separation of kitchen and bathroom | 2.916 | 0.912 |
| | $X_{12}$ | Reliable power supply | 3.839 | 0.842 |
| Public services | $X_{13}$ | Distance to school | 4.080 | 1.024 |
| | $X_{14}$ | Distance to hospitals | 3.534 | 0.864 |
| | $X_{15}$ | Distance to shops | 2.801 | 0.770 |
| | $X_{16}$ | Non-shared toilet | 4.064 | 0.938 |
| | $X_{17}$ | The disposal of the cereal straw | 3.894 | 1.191 |
| Eco-environmental conditions | $X_{18}$ | Safe drinking water | 3.341 | 0.980 |
| | $X_{19}$ | Vegetation coverage | 3.817 | 0.924 |
| | $X_{20}$ | The disposal of the household waste | 3.389 | 1.003 |
| | $X_{21}$ | Pollution by rural enterprise | 2.566 | 0.812 |
| Cultural conditions | $X_{22}$ | Public space for amusement | 2.527 | 0.918 |
| | $X_{23}$ | Employment opportunities | 2.270 | 0.915 |
| | $X_{24}$ | Community security | 2.788 | 0.977 |
| | $X_{25}$ | Public education | 2.778 | 1.077 |
| | $X_{26}$ | Public activities | 2.913 | 0.832 |
| **Endogenous latent variables** | | | | |
| Residents' perceptions | $Y_1$ | Living condition is more capacious | 2.450 | 1.198 |
| | $Y_2$ | Living condition is more convenient | 2.077 | 0.859 |
| | $Y_3$ | Living condition is more cultural | 1.016 | 1.012 |
| | $Y_4$ | Living condition is tidier | 1.450 | 0.400 |

Note: all these factors are perceived by residents, e.g., the complete expression of housing size is that residents' perceptions toward the changes on housing size.

## 4. Results

We use the SEM (Structural Equation Model) to analyze the above survey data, and the final optimal model is depicted below in Figure 2. The Model test results are given in Table 3, and the results of the path analysis are given in Table 4.

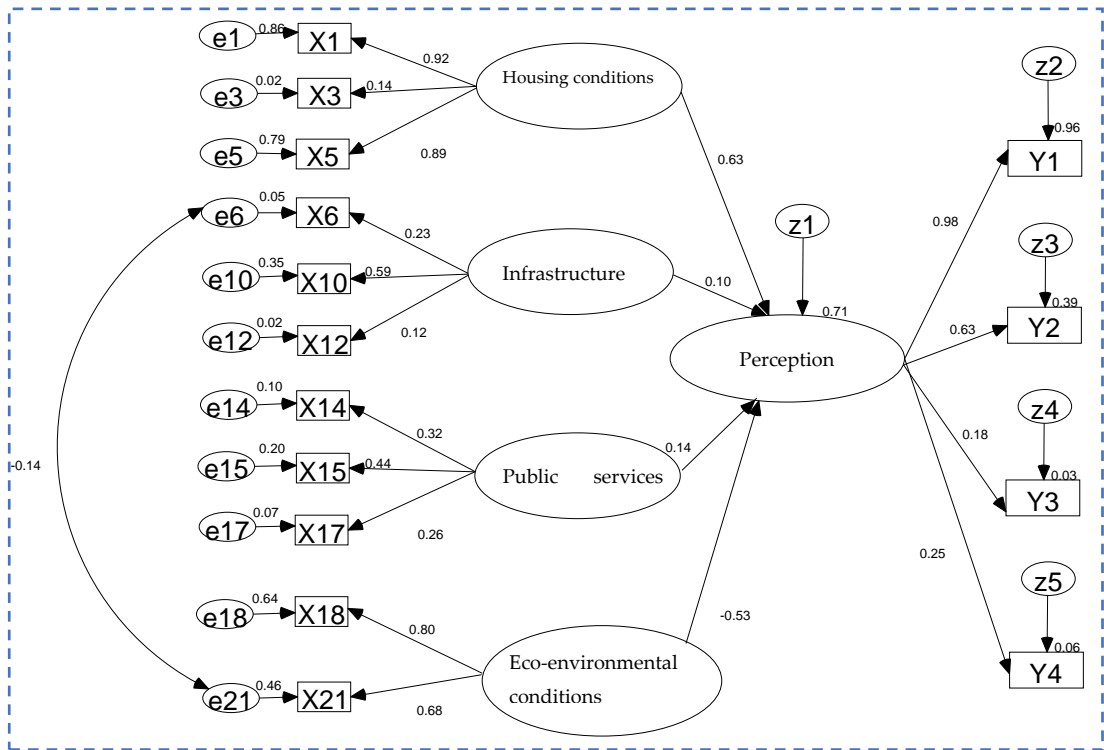

**Figure 2.** The optimal model structure (standardized outcome).

**Table 3.** Evaluation of fitting effect indexes.

| Indexes | Shorthand | Fitted Values | Acceptance Criteria |
|---|---|---|---|
| Absolute fit index | $\chi^2$ | 115.802 | $p > 0.05$ |
| | CMIN/DF | 1.346 | <2.00 |
| | GFI | 0.937 | ≥0.90 |
| | AGFI | 0.902 | ≥0.90 |
| | PGFI | 0.600 | >0.50 |
| | RMSEA | 0.014 | <0.05 |
| | RMR | 0.025 | - - |
| Incremental fit index | NFI | 0.945 | ≥0.90 |
| | CFI | 0.978 | ≥0.90 |
| Parsimonious fit index | AIC | 183.802 | - - |
| | CAIC | 84.955 | - - |
| | CN | 2203 | >200 |

**Table 4.** Results of the path analysis.

| | Path | | | S.E. | C.R. | *p* | STD |
|---|---|---|---|---|---|---|---|
| Resident perceptions | <— | Housing conditions | | — | — | — | 0.631 |
| Resident perceptions | <— | Infrastructure | | 0.219 | 2.940 | *** | 0.103 |
| Resident perceptions | <— | Public services | | 0.317 | 2.460 | *** | 0.139 |
| Resident perceptions | <— | Eco-environmental conditions | | 0.167 | −7.233 | *** | -0.529 |
| $X_1$ | <— | Housing conditions | | — | — | — | 0.925 |
| $X_3$ | <— | Housing conditions | | 0.066 | 2.315 | *** | 0.140 |
| $X_5$ | <— | Housing conditions | | 0.073 | 14.069 | *** | 0.890 |
| $X_6$ | <— | Infrastructure | | — | — | — | 0.231 |
| $X_{10}$ | <— | Infrastructure | | 1.036 | 2.214 | *** | 0.591 |
| $X_{12}$ | <— | Infrastructure | | 0.214 | 2.191 | *** | 0.124 |
| $X_{14}$ | <— | Public services | | 0.522 | 2.365 | *** | 0.323 |
| $X_{15}$ | <— | Public services | | 0.172 | 10.116 | *** | 0.442 |
| $X_{17}$ | <— | Public services | | — | — | — | 0.263 |
| $X_{18}$ | <— | Eco-environmental conditions | | 0.143 | 8.400 | *** | 0.800 |
| $X_{21}$ | <— | Eco-environmental conditions | | — | — | — | 0.680 |
| $Y_1$ | <— | Resident perceptions | | — | — | — | 0.980 |
| $Y_2$ | <— | Resident perceptions | | 0.069 | 5.700 | *** | 0.628 |
| $Y_3$ | <— | Resident perceptions | | 0.060 | 2.100 | *** | 0.183 |
| $Y_4$ | <— | Resident perceptions | | 0.071 | 2.336 | *** | 0.252 |

Notes: *** $p < 0.001$.

### 4.1. Model Test Results

Table 3 below describes the fitness of the model. The goodness of fit index (GFI) and the adjusted goodness of fit index (AGFI) are both >0.90, the parsimony goodness of fit index (PGFI) is >0.50, the root mean square error of approximation (RMSEA) is less than 0.05, and the root mean square residual (RMR) is 0.025, indicating that the model is a suitable fit. In incremental fit indexes, the normed fit index (NFI) and comparative fit index (CFI) are strictly greater than 0.90, meaning that the fitting result is good. In parsimonious fit indexes, the Akaike information criterion (AIC) is 183.802 and the consistent Akaike information criterion (CAIC) is 84.955. There is no uniform standard for AIC and CAIC until now; studies have proven that the smaller the values of AIC and CAIC, the better [38]. Thus, the model indicates good fitness.

### 4.2. The Final Optimal Model

From the perception of rural poverty households, the impact of targeted poverty alleviation on rural living condition is mainly evident in improvement of housing conditions, infrastructure and public services, eco-environmental conditions have obviously weakened. Targeted poverty alleviation has no significant effect on cultural conditions. The final optimal model is depicted below in Figure 2.

### 4.3. Path Analysis

Results of the path analysis are given in Table 4.

TPA shows the most significant improvement effects in the housing conditions of RLCs. There are some improvements, albeit slight, in public services and infrastructure. With regard to the perceptions of rural poverty household (RPHs) during TPA, living conditions are now more capacious ($Y_1$) and living conditions are more convenient ($Y_2$). Here, the degree of reflection reached 0.98 on $Y_1$, demonstrating that residents think that TPA has made living conditions more capacious.

Housing size ($X_1$), housing type ($X_3$), and housing landscape ($X_5$) are three significant factors in housing conditions, in which housing size ($X_1$) is most significant, and the standardized path coefficient is 0.92. The minimum required per-capita usable floor area during the TPA process is 15–20 m$^2$ per capita. At present, the main housing style is a brick-wood construction, so the per-capita usable floor area of RPHs is larger and house styles are more beautiful and modern. Moreover, village governments

have carried out extensive renovations of dilapidated houses, where damaged walls are reconstructed and sometimes modified into storage rooms. We observe that housing conditions are indeed improved.

Water supply ($X_6$), accessibility ($X_{10}$), and reliable power supply ($X_{12}$) are three significant factors in infrastructure, in which accessibility ($X_{10}$) is the most significant, with a standardized path coefficient at 0.59. Running water is supplied via the new Water Diversion from the Yellow River project. In the meantime, village governments have introduced systems for monitoring water safety, improving drinking water, and ensuring accessibility of safe water supply.

It is not the exception, but rather the rule, that rural roads in China are much worse than those of the United States. In the United States, nearly three-quarters of poverty households own a car, and 30% own two or more cars in 2004 [39]. During the TPA process, rural highways in China have been largely improved using sandstone, and traffic safety continues to be promoted, making it more convenient for people. Tricycles are common in the countryside to make life a bit easier for farmers; better roads make the riding faster and safer. Accessibility has also been enhanced. It is also important to mention that by upgrading power grids, power supplies have improved and more broadband connectivity is now installed [40]. Thus, many RPHs have access to broadband internet at home, and this access has much impact on employment.

Distance to hospitals ($X_{14}$), distance to shops ($X_{15}$), and the disposal of cereal straw ($X_{17}$) are three significant factors in the public services category, in which distance to shops ($X_{15}$) is most significant. The standardized path coefficient is 0.44. On the one hand, distance to hospitals has been reduced due to village governments devoting a lot of time to the construction of back roads. Rural medical and health services have developed rapidly as the number of medical service institutions and village doctors increased, and comprehensive health services have become available to rural poverty households. On the other hand, rural express deliveries increasingly provide convenience shopping for consumers in rural areas and the distance to shops is reduced. The Poverty Alleviation and Assistance (PAA) project launched by Lanzhou University in June 2017 found that even before the year 2000, RPHs could only go to the fairs at fixed times (approximately every 10 days per month). In the meantime, people are still too poor to buy enough supplies for daily use; this is in line with the results provided by Jing and Ozanne [41]. Village governments have helped to build express service points for rural poverty households since 2013. As a result, street malls are now more available for residents. Despite the improvements made during the TPA process, the public services in Chedao town are still struggling. In fact, public services are far from perfect in present-day rural China [42]. Potential barriers to public services locating in villages include a lack of public transport, too few laborers, isolation from other agency facilities, and insufficient infrastructure for high-speed telecommunications, and we can confirm similar problems with level of access in our study area.

Safe drinking water ($X_{18}$) and pollution caused by rural enterprises ($X_{21}$) are two significant factors in eco-environmental conditions, in which safe drinking water ($X_{18}$) is the most significant, with its standardized path coefficient being 0.80. Village governments try to meet the demand for drinking water by diverting water from the Yellow River. However, some poor households are not at all satisfied with the actual water quality available. Impure taste, insufficient water supply, and supply interruptions without warning are the major reasons for their dissatisfaction. An more serious issue is that pollution by rural enterprises is further exacerbating the problem of providing clean water. Since one of the main aims in the TPA process is to develop a collective economy, agricultural production co-operatives are significant in the economy [36]. In addition to water pollution concerns, during the construction of cooperatives it is believed that a quiet and pristine environment is violated when rural administrations strongly promote a cooperative economy. Noise pollution is a main negative externality of construction. RPHs mention that eco-environmental conditions are not as environmentally friendly as they used to be because of the noise pollution from rural enterprises.

*4.4. Residents' Perceptions of RLC Change among Various Groups*

The major idea behind the cultural approach is that people are socialized in a culture to share certain values, goals, and behaviors [43]. Similar populations, employment structures, and agrarian conditions are ways in which residents have certain commonalities with regard to the RLC change. At the same time, differences in living conditions have also determined their perceptions about changes in RLC.

While significant improvements in the infrastructure and public services during the TPA occurred, we also note that eco-environmental conditions have significantly weakened (see Figure 2). Villages in Chedao town have been divided into villages of mild rurality, moderate rurality, and severe rurality (see Figure 1). Residents in these different villages have different perceptions of the RLC changes. In the following section, we discuss the different perceptions of residents with regard to changes in different villages.

In the results of our analysis, we can see that housing size ($X_1$), accessibility ($X_{10}$), distance to shops ($X_{15}$), and safe drinking water ($X_{18}$) are the most significant factors for housing conditions, infrastructure, public services, and eco-environmental conditions, respectively. The four aspects of living conditions of different groups are analyzed via key indexes here.

When we examine Figure 3, we can readily observe that residents in different villages differ greatly in their perceptions of rural living condition changes. Poverty households in severe rurality villages have a stronger positive perception of the rural living condition changes during the targeted poverty alleviation (TPA) process compared to poverty households in mild and moderate rurality villages. This means the living conditions before have a significant influence on residents' perceptions of rural living condition change [36]. Since rural poverty households in severe rurality villages are situated far from town centers and due to their poverty, they do not have enough money to transport raw materials for building a house, or to somehow improve their living conditions. Residents in severe rurality villages lack a reliable transport system and live in conditions without running water; these villagers perceive the changes in rural living conditions during the TPA process to be reflected in improved housing conditions.

Housing conditions experienced the most significant change for poverty households in severe rurality villages. Accessibility was most significantly improved factor for poverty households in moderate rurality villages. Distance to shops was markedly improved for poverty households in mild rurality villages. However, importantly, safe drinking water had the lowest improvement in various degrees across all households during the TPA process. Generally, the rural living conditions for rural poverty households in mild rurality villages is as bad as for those living in severe rurality villages. People in mild rurality villages near oil fields and coal mines have the opportunity to obtain part-time jobs. The road network is also in an excellent condition in these villages.

After having accounted for residents' perceptions of the changes brought about by TPA in poverty-stricken villages in NWC, we conclude that housing conditions had the most improvement in poverty households in severe rurality villages; infrastructure had the most improvement in poverty households in moderate rurality villages; and public services are most improved in mild rurality villages. However, eco-environmental conditions obtained the lowest improvement across the board.

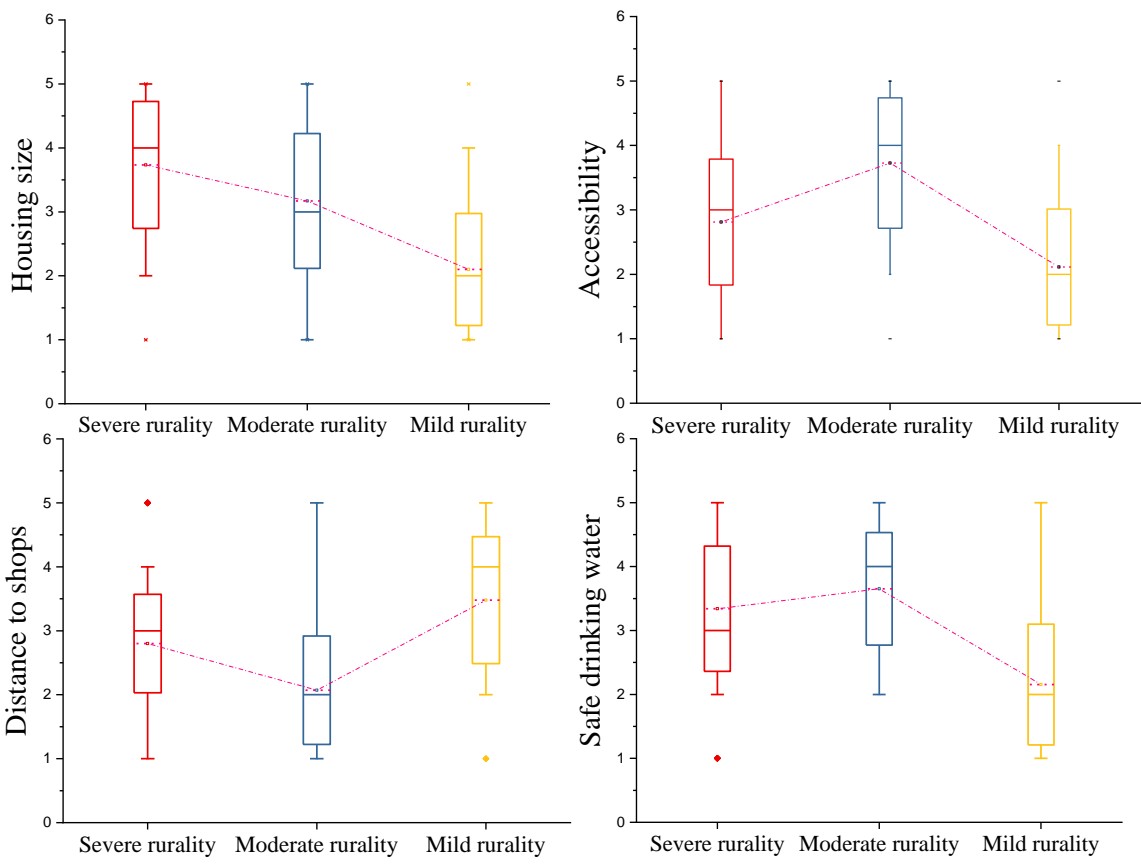

**Figure 3.** Differences of perceptions on changes in RLC in different villages.

## 5. Concluding Remarks: Towards a Better Future for Rural Living Conditions?

The Chinese economy is characterized by a remarkable rural-urban divide [44], as remote villages show significant demographic, infrastructural, and economic differences compared to villages and cities near economic and political centers [36]. The results of the present study of 16 isolated villages of Chedao town, Gansu province, China have clearly demonstrated that, according to residents' perceptions, rural living conditions have changed during targeted poverty alleviation. The main findings indicate that both perceptions of residents and actual improvements were the strongest in the improvement of housing conditions, infrastructure, and public services. Eco-environmental conditions have largely been weakened, however, and targeted poverty alleviation has shown no significant effect on cultural conditions.

Of the factors at stake, our study has found that housing size, accessibility, distance to shops, and safe drinking water are the most significant in housing conditions, infrastructure, public services and eco-environmental conditions, respectively. Generally speaking, all of these 16 villages are extremely poor and inaccessible. Nevertheless, the degree of poverty varies considerably in terms of the geographical distributions of these villages. After having accounted for residents' perceptions of the changes brought about by TPA in poverty-stricken villages in NWC, we conclude that housing conditions obtained the most improvement for poverty households in severe rurality villages; infrastructure had the most improvement in poverty households in moderate rurality villages; and public services experienced the most improvement in mild rurality villages. Also, eco-environmental conditions received the lowest improvements across the board. For the villages with severe rurality, the results do not imply that the infrastructure, public services or the cultural conditions are good, rather, it means that housing conditions are what they are mostly concerned about. In the villages with mild rurality, residents pay more attention on public services because the housing conditions are generally decent. It is obvious that residents in different villages hold different perceptions on change

of rural living conditions. This indicates that local government needs to use more targeted policies for rural living conditions in different villages.

It is the fervent expectation of the people that residents have clear, clean water and a blue sky, and that future generations have green mountains and fertile land. Livable conditions are essential to all residents' well-being; the changes in rural living condition of rural poverty households have focused on the welfare of poverty households, improvement of social harmony, and rural development in the targeted poverty alleviation process. Governments at all levels could take steps to strengthen rural living conditions in the following ways.

The improvement of infrastructure and public services is fundamentally important in both severe rurality villages and moderate rurality villages. Village governments should optimize infrastructure and public services with close attention to the different types of geography and circumstances of each village, as well as focusing on the different locations of each village [45]. Key points of improvement of rural living conditions in different types of villages should be addressed. Village governments in severe rurality villages would benefit by incorporating the results of the targeted poverty alleviation into planning procedures. In mild rurality villages, the highest priority instead is for governments to build more cultural facilities, give villagers 'on demand' access to radio and film, promote farmers' recreation, and increase the ownership of phones and computers [46]. Lastly, the improvement of eco-environmental conditions across all villages is sorely needed in NWC. Environmental improvements such as clean and tidy villages, continuous improvements in the livability index, renovation of dilapidated rural houses, greater rural living security, and the ability to prevent and mitigate disasters are some of the most necessary tasks for village governments.

Chen and Uitto (2015) have argued that the government of China faces major challenges in providing environmental protection services, without the mature participation of civil society [47]. We do find a similar condition here in our case studies. However, as what we have demonstrated in this article, it is rather difficult for the local residents to improve their livelihoods and living environment by themselves because of the poor natural conditions and isolation of the whole town. In fact, there is hardly any easy way to promote rural living conditions for rural poverty households in remote rural areas in northwest China. The improvements of infrastructure still need support from the government. It can be expected that when the basic needs of their village residents have been fulfilled, they will be endowed with more initiative to improve their livelihoods more by themselves rather than by the government.

**Author Contributions:** M.W. came up with an idea of R.L. Conditions change by Targeted Poverty Alleviation Policy, then did field research and wrote the whole manuscript. Y.Y. refined the structure of the article, B.Z. enriched article contents, M.L. made figures, and Q.L. participated in the revision of the article.

**Funding:** This article is financially supported by the Fundamental Research Funds for the Central Universities of China (No. lzujbky-2019-31), the National Natural Science Foundation of China (No. 41971198), Natural Science Foundation of Guangdong Province, China (No. 2019A1515011385), and National social science fund project of China (No. 16CGL032).

**Acknowledgments:** The authors would like to express their sincere thanks to the 2203 rural poverty households for accepting our interview, many people who give us advice, and our reviewer.

**Conflicts of Interest:** The author(s) declared no potential conflicts of interests with respect to the authorship and/or publication of this article.

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
