# Peer review of "How Does Targeted Poverty Alleviation Policy Influence Residents’ Perceptions of Rural Living Conditions? A Study of 16 Villages in Gansu Province, Northwest China"

_sustainability, doi:10.3390/su11246944_

Round 1

Reviewer 1 Report

This is an empirical study in Gansu Province. The topic – poverty alleviation in rural China – is obviously very important and the study provides some interesting findings. I also found that the perspective by this group of researchers is refreshingly frank and critical, yet strongly evidence-based. Beginning the article with the story about the farmer Yang’s family places the research into a gripping context. The research design appears appropriate and is adequately described. The results are presented systematically and clearly. My comments thus focus on where I believe some improvements could be made.

Section 2. Literature Review is generally good. I was very pleased with the emphasis placed on multidimensional poverty. I do think it would be useful to explicitly mention UNDP’s work on the MPI (hdr.undp.org/en/2019-MPI). It is good that the authors have also identified researchers who have not used MPI but have highlighted the multidimensional nature of poverty in other ways. On the MPI itself, the paper cites fundamental work by Alkire and her associates, as well as Zhi et al. (2017) and the Syria study by Rovere et al. (2006), the latter of which appears a bit random and the list could be supplemented by other references. For example, Rogan (2016) highlights the missing gender dimension. Another possible study to cite would be Angulo et al. (2016).

Section 5. Concluding Remarks is good and concise. Based on the evidence gathered, the authors duly highlight what has worked and what has not, drawing important attention to the “eco-environmental conditions.” I also particularly like their emphasis on aspects that tend to get less attention, notably the importance of cultural facilities. My only comment here is that the recommendations are all directed to different levels of government, as if only government could improve the situation. In fact, it has been argued (Chen and Uitto, 2015) that there is an over-reliance on government services in China and the government, too, would do better to delegate more authority to civil society.

As for the presentation, it would be useful to have another map in addition to Figure 1, one that would show the location of the study area in Gansu and in China.

With regard to the language, the paper is in good shape and is easy to read. Still, the final version could use a native English-speaker to read through. Furthermore, the authors use many abbreviations unnecessarily, especially when a term is not a standard one or when it is not mentioned that many times. Sentences like, “The RLC of RPHs has witnessed great changes in the TPA…” (p. 3) don’t at all make the paper easier to read.

References:

Angulo, R., Y. Diaz and R. Pablo. The Colombian multidimensional poverty index: Measuring poverty in a public policy context. Social Indicators Research, 2016, 127 (1): 1-38.

Chen, S. and J.I. Uitto. Accountability delegation: Empowering local communities for environmental protection in China. Development, 2015, 58 (2-3): 354-365.

Rogan, M. Gender and multidimensional poverty in South Africa: Applying the global Multidimensional Poverty Index. Social Indicators Research, 2016, 126 (3): 987-1006.

Author Response

Authors’ Response to the Review Comments

Manuscript: Sustainability-644942

Journal: Sustainability

Title of Paper: How Does Targeted Poverty Alleviation Policy Influence the Residents’ Perception of Rural Living Condition? A Study of 16 Villages in Gansu Province, Northwest China

Date Sent: 2019-11-28

We appreciate the time and efforts by the editor and referees in reviewing this manuscript. We believe we have addressed all issues indicated in the review report; the new version was following closely with the comments from all the reviewers and the requirement of the journal. We hope that the revised version can meet the journal publication requirements.

Reviewer 1:

This is an empirical study in Gansu Province. The topic – poverty alleviation in rural China – is obviously very important and the study provides some interesting findings. I also found that the perspective by this group of researchers is refreshingly frank and critical, yet strongly evidence-based. Beginning the article with the story about the farmer Yang’s family places the research into a gripping context. The research design appears appropriate and is adequately described. The results are presented systematically and clearly. My comments thus focus on where I believe some improvements could be made.

Section 2. Literature Review is generally good. I was very pleased with the emphasis placed on multidimensional poverty. I do think it would be useful to explicitly mention UNDP’s work on the MPI (hdr.undp.org/en/2019-MPI). It is good that the authors have also identified researchers who have not used MPI but have highlighted the multidimensional nature of poverty in other ways. On the MPI itself, the paper cites fundamental work by Alkire and her associates, as well as Zhi et al. (2017) and the Syria study by Rovere et al. (2006), the latter of which appears a bit random and the list could be supplemented by other references. For example, Rogan (2016) highlights the missing gender dimension. Another possible study to cite would be Angulo et al. (2016).

Angulo, R., Y. Diaz and R. Pablo. The Colombian multidimensional poverty index: Measuring poverty in a public policy context. Social Indicators Research, 2016, 127 (1): 1-38.

Rogan, M. Gender and multidimensional poverty in South Africa: Applying the global Multidimensional Poverty Index. Social Indicators Research, 2016, 126 (3): 987-1006.

Response: Thank you very much for the suggestion. We have cited Angulo, R., Y. and Rogan, M. in line 155of page 4 in our new manuscript.

“…Correspondingly, the Decade Poverty Reduction Programme (DPRP 2011) of China proposed improvements to residence, medical services and education. In fact, these dimensions and aspects have been widely discussed in adademia. Angulo et al. (2016) design poverty reduction goals in Colombian through monitoring the public policies, Rogan (2016) highlights the missing gender dimension.…”

Section 5. Concluding Remarks is good and concise. Based on the evidence gathered, the authors duly highlight what has worked and what has not, drawing important attention to the “eco-environmental conditions.” I also particularly like their emphasis on aspects that tend to get less attention, notably the importance of cultural facilities. My only comment here is that the recommendations are all directed to different levels of government, as if only government could improve the situation. In fact, it has been argued (Chen and Uitto, 2015) that there is an over-reliance on government services in China and the government, too, would do better to delegate more authority to civil society.

Chen, S. and J.I. Uitto. Accountability delegation: Empowering local communities for environmental protection in China. Development, 2015, 58 (2-3): 354-365.

Response: We have added the discussion in line 423 of page 13 in our new manuscript.

Chen and Uitto (2015) have argued that the government of China faces the great challenges in providing environmental protection services, without a mature participation of civil society. We do find the similar condition here in our case studies. However, as what we have demonstrated in this article, it is rather difficult for the local residents to improve their livelihoods and living environment by themselves because of the poor natural conditions and isolation of the whole town. In fact, there is hardly any easy way to promote rural living conditions for rural poverty households in remote rural area, in northwest China. The improvements of infrastructure still have to rely on the government. It can be expected that when the basic needs of their village residents have been fulfilled, they will be endowed more initiative to improve their livelihoods more by themselves rather than by government.

As for the presentation, it would be useful to have another map in addition to Figure 1, one that would show the location of the study area in Gansu and in China.

Response: Thank you. We have replaced Figure 1 with a new specific figure.

With regard to the language, the paper is in good shape and is easy to read. Still, the final version could use a native English-speaker to read through. Furthermore, the authors use many abbreviations unnecessarily, especially when a term is not a standard one or when it is not mentioned that many times. Sentences like, “The RLC of RPHs has witnessed great changes in the TPA…” (p. 3) don’t at all make the paper easier to read.

Response: Thank you, it is done.

References:

Angulo, R., Y. Diaz and R. Pablo. The Colombian multidimensional poverty index: Measuring poverty in a public policy context. Social Indicators Research, 2016, 127 (1): 1-38.

Chen, S. and J.I. Uitto. Accountability delegation: Empowering local communities for environmental protection in China. Development, 2015, 58 (2-3): 354-365.

Rogan, M. Gender and multidimensional poverty in South Africa: Applying the global Multidimensional Poverty Index. Social Indicators Research, 2016, 126 (3): 987-1006.

Response: Yes, we added them in our new manuscript.

Reviewer 2 Report

How Does Targeted Poverty Alleviation Policy Influence the Residents’ Perception of Rural Living Condition? A Study of 16 Villages in Gansu Province, Northwest China

This is a well-written paper that tackles relevant issues for better informing poverty alleviation strategies in Northwest China. The literature review is comprehensive, the method appropriate, the empirical investigationis thorough, and the discussion points to key residents’ perceptions on the impact of poverty alleviation measures.  Policy recommendations for the improvement of rural living conditions are formulated according to the different types of villages.

I have nevertheless two suggestion aiming to make more explicit the method applied in this article:  

weights calculation with the variation coefficient method should be either briefly explained or referenced. It is not entirely clear how did the authors breakdown the sample into mild rurality villages, moderate rurality villages and severe rurality villages. What are the distinctive features of each group?

Author Response

Authors’ Response to the Review Comments

Manuscript: Sustainability-644942

Journal: Sustainability

Title of Paper: How Does Targeted Poverty Alleviation Policy Influence the Residents’ Perception of Rural Living Condition? A Study of 16 Villages in Gansu Province, Northwest China

Date Sent: 2019-11-28

We appreciate the time and efforts by the editor and referees in reviewing this manuscript. We believe we have addressed all issues indicated in the review report; the new version was following closely with the comments from all the reviewers and the requirement of the journal. We hope that the revised version can meet the journal publication requirements.

How Does Targeted Poverty Alleviation Policy Influence the Residents’ Perception of Rural Living Condition? A Study of 16 Villages in Gansu Province, Northwest China,This is a well-written paper that tackles relevant issues for better informing poverty alleviation strategies in Northwest China. The literature review is comprehensive, the method appropriate, the empirical investigations thorough, and the discussion points to key residents’ perceptions on the impact of poverty alleviation measures.  Policy recommendations for the improvement of rural living conditions are formulated according to the different types of villages.

I have nevertheless two suggestion aiming to make more explicit the method applied in this article:  weights calculation with the variation coefficient method should be either briefly explained or referenced.

Response: Thank you very much. The literature on algorithms is cited in line 193 of page 4 in our new manuscript. Please see the red part one page 4.

It is not entirely clear how did the authors breakdown the sample into mild rurality villages, moderate rurality villages and severe rurality villages. What are the distinctive features of each group?

Response: We discussed regional division process and the distinctive features of each group in detail, in line 194 of page 5 of our new manuscript.

"…Table 1 sets out the index of rurality indicators, the calculation of weights (Wi) is based on the variation coefficient method (Faber and Korn, 1991). The extreme difference method is used in data standardization and the different directions of indexes are identified. According to index weight and standardized value, the rural indexes of each village are calculated. Villages with similar property values are set as a cluster by ArcGIS, the 16 villages in Chedao town are divided into mild rurality villages, moderate rurality villages and severe rurality villages (See Figure 1). Villages in Chedao town are varying tremendously in terms of rurality. villages with mild and moderate rurality are located in the northeast part of Chedao town where the agricultural productivity is relatively higher and small-scale cooperatives are relatively better established, therefore, fewer people move out for a living. On the contrary, the villages with severe rurality are highly constrained by the shortage of water and low agricultural technology, in other words, the agricultural productivity in these villages is largely depended on the weather condition. This means many people work outside and the villages are facing a labor shortage. The condition in the villages with moderate rurality lies somewhere in between mild rurality villages and severe rurality villages ...”

Yours sincerely,

Meimei Wang, Yongchun Yang, Bo Zhang, Mengqin Liu, and Qing Liu

Nov. 29th, 2019

Reviewer 3 Report

The rural poverty in China is for sure a topic that deserves attention.

Nevertheless, the article does not demonstrate sufficiently the scientific relevance of the qualitative research here presented. Most of all, it is not well-explained the contribution to the existing literature.

Specifically, it is not clear the scope of the paper: is the research proposing an assessment of the present public policy against rural poverty or is it just providing further data from a territorial context scarcely investigated?

And, given the serious condition of the population in Northwest China, why the population's perception of the policies is more relevant than a quantitative assessment of the effects on the population's quality of life? 

Moreover, the conclusions of qualitative research are poor: it is quite obvious that clean water in a polluted area, sufficient housing size, accessibility to transportations and minimum services are essential features for the population.

The authors have to demonstrate the originality and the contribution to the existing literature on the topic.

Author Response

Authors’ Response to the Review Comments

Manuscript: Sustainability-644942

Journal: Sustainability

Title of Paper: How Does Targeted Poverty Alleviation Policy Influence the Residents’ Perception of Rural Living Condition? A Study of 16 Villages in Gansu Province, Northwest China

Date Sent: 2019-11-28

We appreciate the time and efforts by the editor and referees in reviewing this manuscript. We believe we have addressed all issues indicated in the review report; the new version was following closely with the comments from all the reviewers and the requirement of the journal. We hope that the revised version can meet the journal publication requirements.

Nevertheless, the article does not demonstrate sufficiently the scientific relevance of the qualitative research here presented. Most of all, it is not well-explained the contribution to the existing literature.

Response: Thank you very much. The relevant useful literature has been added in line 111 of page 4 of our new manuscript.

“… The contributions of this article lie threefold: 1. We focus on the area where scarcely research has ever studied before because of the low accessibility for the scholars. 2. The existing literature mostly concentrates on how the improvement of infrastructure has influenced the livelihoods of the residents, relatively less research pays attention to how the rural residents feel and experience after the changes of the rural living condition. Specifically, in this article, we stress on the perception of the residents. 3. Although this study is conducted in a small scale, we find the degree of poverty still varies considerably in terms of geographical distributions of these villages. The results indicate that the local government does need to make effort to the targeted poverty alleviation on the rural livelihoods of the residents due to the geographical distributions ...”

Specifically, it is not clear the scope of the paper: is the research proposing an assessment of the present public policy against rural poverty or is it just providing further data from a territorial context scarcely investigated? And, given the serious condition of the population in Northwest China, why the population's perception of the policies is more relevant than a quantitative assessment of the effects on the population's quality of life? 

Response: Thank you. In this article, we would like to focus on the role of present public policy against rural poverty in remote areas and make assessment of the changes after the poverty alleviation project based on the perceptions of local resident . We discussed this issue in line 100 of page 3 of our new manuscript.

“…It seems that the infrastructures are improved during our several visits to Chedao town, it is nevertheless still unknown how exactly the residents feel about these improvements. The perceptions of the residents can be important indicators to assess the present public policy against rural poverty …”

Moreover, the conclusions of qualitative research are poor: it is quite obvious that clean water in a polluted area, sufficient housing size, accessibility to transportations and minimum services are essential features for the population. The authors have to demonstrate the originality and the contribution to the existing literature on the topic.

Response: We discussed the conclusion and possible contribution of our article in detail, in line 389 of page 12 of our new manuscript.

“…16 villages are extremely poor and inaccessible. Nevertheless, the degree of poverty varies considerably in terms of geographical distributions of these villages. After having accounted for residents’ perceptions of the changes brought about by TPA in poverty-stricken villages in NWC, we conclude that housing conditions get the most improvement in poverty households in severe rurality villages; infrastructure gets the most improvement in poverty households in moderate rurality villages; and public services are most improved in mild rurality villages. And eco-environmental conditions receive the lowest improvement across the board. For the villages with severe rurality, the results do not imply that the infrastructure, public services or the cultural conditions are good, rather, it means that housing conditions are what they are mostly concerned. In the villages with mild rurality, residents pay more attention on public services because the housing conditions are generally OK. It is obviously that residents in different villages hold different perceptions on change of rural living condition. This indicates that local government needs to make more targeted policies on rural living condition in different villages …”

Yours sincerely,

Meimei Wang, Yongchun Yang, Bo Zhang, Mengqin Liu, and Qing Liu

Nov. 29th, 2019

Round 2

Reviewer 3 Report

The authors addressed all the comments properly, and the overall quality of the article is certainly improved.